# Pediatric Neuroendocrine Neoplasia of the Parathyroid Glands: Delving into Primary Hyperparathyroidism

**DOI:** 10.3390/biomedicines11102810

**Published:** 2023-10-17

**Authors:** Mara Carsote, Mihaela Stanciu, Florina Ligia Popa, Ana-Maria Gheorghe, Adrian Ciuche, Claudiu Nistor

**Affiliations:** 1Department of Endocrinology, Carol Davila University of Medicine and Pharmacy, 050474 Bucharest, Romania; carsote_m@hotmail.com; 2C.I. Parhon National Institute of Endocrinology, 011863 Bucharest, Romania; anamaria.gheorghe96@yahoo.com; 3Department of Endocrinology, Faculty of Medicine, “Lucian Blaga” University of Sibiu, Victoriei Blvd., 550024 Sibiu, Romania; 4Department of Physical Medicine and Rehabilitation, Faculty of Medicine, “Lucian Blaga” University of Sibiu, Victoriei Blvd., 550024 Sibiu, Romania; 5Ph.D. Doctoral School, Carol Davila University of Medicine and Pharmacy, 050474 Bucharest, Romania; 6Department 4—Cardio-Thoracic Pathology, Thoracic Surgery II Discipline, Carol Davila University of Medicine and Pharmacy, 050474 Bucharest, Romania; adrianciuche@gmail.com (A.C.); ncd58@yahoo.com (C.N.); 7Thoracic Surgery Department, Dr. Carol Davila Central Emergency University Military Hospital, 010825 Bucharest, Romania

**Keywords:** neuroendocrine, primary hyperparathyroidism, multiple endocrine neoplasia, surgery, pediatric, parathyroidectomy, ectopic, calcium, PTH, thoracoscopy

## Abstract

Our objective was to overview the most recent data on primary hyperparathyroidism (PHP) in children and teenagers from a multidisciplinary perspective. Methods: narrative review based on full-length, English-language papers (from PubMed, between January 2020 and July 2023). Results: 48 papers (14 studies of ≥10 subjects/study, and 34 case reports/series of <10 patients/study). Study-sample-based analysis: except for one case–control study, all of the studies were retrospective, representing both multicenter (n = 5) and single-center (n = 7) studies, and cohort sizes varied from small (N = 10 to 19), to medium-sized (N = 23 to 36) and large (N = 63 to 83); in total, the reviewed studies covered 493 individuals with PHP. Case reports/series (n = 34, N = 41): the mean ages studied varied from 10.2 to 14 years in case reports, and the mean age was 17 years in case series. No clear female predominance was identified, unlike that observed in the adult population. Concerning the assessments, there were four major types of endpoints: imaging data collection, such as ultrasound, 99mTc Sestamibi, or dual-phase computed tomography (CT); gene testing/familial syndrome identification; preoperative findings; and exposure to surgical outcome/preoperative drugs, like cinacalcet, over a 2.2-year median (plus two case reports of denosumab used as an off-label calcium-lowering agent). Single-gland cases (representing 85% of sporadic cases and 19% of genetic PHP cases) showed 100% sensitivity for neck ultrasounds, with 98% concordance with 99mTc Sestamibi, as well as a 91% sensitivity for dual-phase CT, with 25% of the lesions being ectopic parathyroids (mostly mediastinal intra-thymic). Case reports included another 9/41 patients with ectopic parathyroid adenomas, 3/41 with parathyroid carcinomas, and 8/41 subjects with brown tumors. Genetic PHP (which has a prevalence of 5–26.9%) mostly involved *MEN1*, followed by *CDC73*, *CASR*, *RET*, and *CDKN1B*, as well as one case of *VHL*. Symptomatic PHP: 70–100% of all cases. Asymptomatic PHP: 60% of genetic PHP cases. Renal involvement: 10.5% of a cohort with genetic PHP, 71% of sporadic PHP cases; 50% (in a cohort with a mean age of 16.7), 29% (in a cohort with a mean age of 15.2); 0% (in infancy) to 50–62% (in teenagers). Bone anomalies: 83% of the children in one study and 62% of those in two other studies. Gastrointestinal issues: 40% of one cohort, but the data are heterogeneous. Cure rate through parathyroidectomy: 97–98%. Recurrent PHP: 2% of sporadic PHP cases and 38% of familial PHP cases. Hungry bone syndrome: maximum rate of 34–40%. Case reports identified another 7/41 subjects with the same post-parathyroidectomy condition; a potential connection with ectopic presentation or brown tumors is suggested, but there are limited data. Minimally invasive thoracoscopic approaches for ectopic tumors seemed safe. The current level of statistical evidence on pediatric PHP qualifies our study- and case-sample-based analysis (n = 48, N = 534) as one of the largest of its kind. Awareness of PHP is the key factor to benefit our young patients.

## 1. Introduction

Pediatric primary hyperparathyroidism (PHP) represents a rare condition that is usually caused by a parathyroid neuroendocrine tumor, most commonly an adenoma (in 65–95% of cases) of small dimensions. Since clinical evaluation of the tumor itself is rather irrelevant, the index of suspicion usually comes from acute and chronic hypercalcemia-related complications [1,2]. Imaging tools are essential to locating the parathyroid tumor, starting with a traditional anterior cervical ultrasound, up to the use of specific functional imaging, such as the use of 99 m Technetium (Tc) Sestamibi scintigraphy, SPECT/CT (single-photon emission computerized tomography/computed tomography), 4D-CT, 4D-resonance magnetic imaging, etc. [3,4,5,6]. Surgical approaches, similar to those seen in adult populations, are the only methods that provide a disease cure [7,8,9]. Recently, thoracoscopic resection of ectopic adenomas (that generally represent 6–16% of all parathyroid tumors) has been applied in young patients, as well [10,11,12].

As opposed to cases of adults admitted for the traditional panel of PHP (not the modern picture of asymptomatic PHP), PHP-associated clues are unusual findings among infants and teenagers, due to a reduced epidemiological impact (with an incidence of 1 case/300,000 children, or up to 2–5 cases/100,000, still almost 100 times less frequent than PHP in adults); thus, diagnosis may be delayed, and the presentation may be complicated by multiple bone, renal, digestive, cardiac, and/or growth disturbances [13,14,15].

A genetic background is found in 10–30% of juvenile cases, typically involving deletions/duplications or pathogenic variants of the *MEN1*, which underlines multiple endocrine neoplasia type 1 syndrome (MEN1), *CDC73* (also known as *HRPT2*; its germline pathogenic variant causes hyperparathyroidism (jaw tumor syndrome)), *CDKN1B*, *RET* (MEN2), *AIP*, *CDKN1B*, and *CASR* (calcium-sensing receptor) genes, requiring differential diagnosis of familial hypocalciuric hypercalcemia, which might be complicated by severe neonatal PHP with homozygote status [16,17,18,19]. In adults, one out of ten subjects confirmed with PHP has a genetic condition, most commonly MEN1, MEN4, MEN2, hereditary hyperparathyroidism (jaw tumor syndrome), or isolated familial PHP [20,21,22].

Overall, pediatric PHP is considered more severe when compared to PHP in adults, due to its heterogeneous presentation, various complaints that a child cannot entirely describe, and its rarity, which leads to a more challenging index of suspicion; also, a complex pathogenic panel, with regard to mineral and bone metabolism changes during growth and puberty, including the achievement of peak bone mass, could potentially be affected by the high calcium and/or PTH (parathyroid hormone) levels caused by the condition. For these reasons, it is mandatory to address this topic from a modern multidisciplinary perspective, in order to increase awareness around it and to facilitate its early recognition, thus reducing the disease burden, a result which has proven to be life-saving in certain patients [2,23,24,25].

### Aim

Our objective was to overview the most recent data concerning neuroendocrine tumors causing PHP in children and teenagers from the multidisciplinary perspectives of pathogenic traits, clinical presentations, genetic analysis, management, and outcomes. 

## 2. Materials and Methods

For this narrative review, we searched for full-length papers, published in the English language, that were freely accessed via PubMed according to the key words “pediatric primary hyperparathyroidism”, “child and primary hyperparathyroidism”, and “teenager and primary hyperparathyroidism” (Table 1).

According to the three PubMed search strategies within the mentioned timeframe, we identified 92, 115, and 95 papers, respectively, then applied the exclusion criteria, removed duplicates, and ultimately included the following in this 3-year case- and study-sample-based analysis: 48 original articles (14 studies (≥10 patients/study), 27 individual case reports without, and 7 case reports and series (<10 patients/article)with, specific genetic testing (n = 34 case reports/series)) (Figure 1).

## 3. An Update of Pediatric Primary Hyperparathyroidism: From Admission to Outcome 

### 3.1. Clinical Presentation

The clinical elements may be acute (with recent onset), or there may be a history of a few months or even years of various complications, including impairment in achieving normal growth and pubertal development in severe cases; some authors consider the clinical picture to be more severe than that seen in adults: children with genetic PHP who come from families with a known mutation may be the subjects of surveillance programs; thus, the clinical picture may be deficient [24,26]. One study from 2022, on 66 pediatric patients with sporadic and familial PHP with a mean age of 17 years (studied between 1994 and 2020), showed that the clinical presentations were suggestive for the diagnosis of PHP in 71% of the subjects (symptomatic PHP) [27]. A retrospective juvenile study (N = 35; mean age of 15.2 years; male to female ratio of 1 to 1.9) revealed symptomatic PHP in 94% of the cohort; the most frequent clinical presentation was related to bone disturbances (83%), followed by renal complications in 29% of the children and teenagers. Fractures were confirmed in 54% of the individuals, while hypercalcemic crisis was identified in 2.8% of them, and recurrent pancreatitis in 11.4% [28]. 

Bin Yahib et al. [29], while reporting on a teenager with recurrent abdominal pain as a dominant feature on admission, performed a sample case analysis between 2007 and 2020 (13 articles regarding a total of 331 individuals diagnosed with PHP, aged 22 years and younger) and identified the main symptoms of the clinical presentation as being renal (like kidney stones and polyuria) in 27.41% of the patients, osseous (bone pain, deformities, and fractures) in 23.49% of the individuals, fatigue-related (and lethargy) in 20.48% of the cohort, gastrointestinal (for instance, recurrent abdominal pain, anorexia, nausea, vomiting, weight loss, and jaundice) in 18.67% of the subjects, and neurological (for example, memory loss, incapacity to concentrate, and depressive mood) in 17.46% of them. On the other hand, the incidental diagnosis of PHP was established in 5.44% of the cases, and asymptomatic PHP was confirmed in 3.3% of the 331 patients [29]. Abdominal pain may affect 11–45% of the pediatric population, while 29% to 87% of juvenile cases with PHP involve this complaint [29,30,31,32,33]. Of note, abdominal pain may be caused by hypercalcemia-induced pancreatitis or PHP-related kidney stones (sometimes accompanied by hematuria) [34,35]; these should not be missed due to the fact that a child with PHP might present a certain level of emotional instability, chronic asthenia, or other aches, such as bone pain [36].

Respiratory complaints are rarely reported; for instance, an 18-year-old female was admitted for chest pain and dyspnea, and she was diagnosed with non-syndromic PHP, complicated with an osteolytic rib lesion. She underwent left inferior parathyroidectomy, which was followed by hungry bone syndrome; the pathological report pointed out a parathyroid tumor with cystic transformation [15]. 

Rickets may be coincidental, and sometimes this delays the recognition of PHP. For example, a 12-year-old boy coming from Sudan presented with a 6-month history of bone pain, loss of appetite, and fatigue, associated with progressive lower limb deformity that impaired normal walking. Left inferior parathyroid adenoma-associated PHP was masked by rickets anomalies, such as genu valgum that required further surgical correction after successful parathyroid tumor removal. Assessing the status of the vitamin D level helps with the decision of vitamin D supplementation, including in selective cases, with PHP-associated hypercalcemia [37]. However, Dikova et al. [38] reported two teenagers with PHP-associated genu valgum [38]. Another case with PHP-related genu valgum was published by Lee et al. [39]. A teenager suffering from bilateral genu valgum and associated short stature was referred for orthopedic correction at the age of 15 (with normal calcium values at that point); however, 2 years later, he suffered multiple fractures due to adenoma-associated PHP causing brown tumors. A good outcome was registered after parathyroidectomy [40]. In 2020, Lee et al. [39] reviewed all the cases with genu valgum and PHP reported between 1945 and 2019 and found 23 cases. The potential pathogenic mechanism is represented by PTH over excess and its contributions to the epiphyseal plate and to the bone remodeling amid puberty; however, this knowledge is still incomplete [39,41].

Giant parathyroid tumors are exceptional in children; they may be associated with a more severe clinical picture compared to small-sized parathyroid adenomas. A 16-year-old male was admitted for severe digestive complaints, caused by biochemical and hormonal issues related to a giant parathyroid adenoma (that was large enough to be palpable) [9]. Interestingly, this patient also displayed cerebral calcification at the level of frontal lobe (according to a CT scan) that was considered by Muse et al. [9] to be the first reported case in pediatric PHP (in 2023) complicated with this unusual entity [9]. While central calcifications are rather signature of hypoparathyroidism, they have been exclusively reported in adults with PHP, and they do not belong to the classical picture of PHP, as opposed to nephrolithiasis or renal calcifications of any size [42,43,44].

Another unusual neurological presentation of PHP was reported by Pal et al. [44], namely, posterior reversible encephalopathy syndrome due to hypercalcemia (manifested as generalized seizures and anomalies of the sensorium). These two patients represent the first of their kind in the pediatric population according to the authors, who identified only four other cases, all in adults [44] (Table 2).

### 3.2. Genetic Considerations

MEN1-associated PHP represents a major part of the genetic picture accompanying PHP affecting both children and adults. Some controversies around the use of total/partial parathyroidectomy and their timing still exist, as patients may experience other tumors, such as gastrointestinal, lung, thymic, etc., and pituitary neuroendocrine neoplasia, leading to a dramatic burden and a reduced quality of life [53,54]. Screening protocols start in the second decade of life, and MEN1-related pediatric PHP seems the most important gene-associated form of PHP [55,56,57]. MEN1, an autosomal dominant condition with a clinical onset before the age of 20, is not studied across generous published cohorts with respect to juvenile PHP, and we only have a limited amount of information so far on this specific matter [58,59,60]. 

Shariq et al. [49] published a retrospective study on 80 individuals with MEN1 who began screenings before the age of 18. Of them, 70% had at least one tumor confirmed by this cutoff age (at a median age of 14), while PHP was identified in 80% of them (>70% of this subgroup already had a parathyroidectomy performed); 35% of the patients had a pancreaticoduodenal neuroendocrine tumor, and 30% had pituitary neuroendocrine neoplasia [49]. 

Interestingly, in 2021, Srirangam Nadhamuni et al. [61] reported the first pediatric case of MEN1-associated gigantism due to ectopic Growth-Hormone-Releasing Hormone (GHRH) production, in an 18-year-old female who also underwent parathyroidectomy for PHP at the age of 15 [61]. Moreover, a new *MEN1* germline pathogenic variant was described in 2020 (heterozygous variant c.105_107dupGCT) starting from a proband who was a teenager diagnosed with PHP and a pituitary micro-adenoma; further confirmation of a similar phenotype–genotype combination was conducted in her father [62]. 

If MEN1-associated PHP is detected in a child (as an index case), first degree relatives should be tested, as well, and start surveillance protocols if they also harbor the pathogenic variant [63]. Petriczko et al. [64] reported the case of a 17-year-old male who presented with a pancreatic tumor in association with MEN1-related PHP; his sister had the same tumors, but also an unusual central ganglioglioma; they inherited the condition from their father, who was also confirmed with PHP [64].

Blackburn et al. [17] also introduced a case series (N = 2) regarding *CDC73* (*cell division cycle*) pathogenic variants and pediatric PHP, with patients of 10 and 14 years old having symptomatic hypercalcemia [17]. On the other hand, Ramonell et al. [47] identified a case of MEN2–associated PHP harboring *RET* mutations, which seem rarer than previously mentioned two, *MEN1* and *CDC73* [47]. Infants are more likely to be affected by *CASR* mutations, which, in heterozygote status, do not present as classical PHP. One study on 63 pediatric patients with PHP (from 1998 to 2018) showed that 52% of them had a genetic condition; while infants were more likely to be asymptomatic when compared to older children (15% vs. 54%, *p* = 0.002), the younger group were associated with pathogenic variants, particularly (94%) involving *CASR*, whereas the older group presented other mutations, such as *MEN1*, *CDC73*, *RET*, and *CDKN1*, which were associated with higher PTH levels, but with similar serum calcium values to the infant group [50]. 

The reclassification from sporadic to genetic (familial) PHP stands out as a pitfall of juvenile PHP. Szabo Yamashita T. et al. [27] showed that, among 66 pediatric patients with PHP, approximately one-third of them had a prior-known familial syndrome (mostly *MEN1)*, and another 5% of the children initially considered as having a sporadic PHP turned out to have a genetic form. The study also provided insightful information concerning the presentations and outcomes in sporadic vs. familial PHP; for instance, the rates of single-parathyroid-gland involvement were 85% vs. 19%, respectively; postoperative recurrent PHP was identified in 2% vs. 38% of patients, respectively, with a time to recurrence higher in sporadic PHP [27]. Similarly, genetic/familial PHP vs. apparently sporadic PHP was studied in a single-center analysis, published in 2022 by Sharma et al. [26] (N = 36 Asian/Indian patients younger than 20 years; 55% were males; the median age was 17 years). Of the subjects from the first group, 90% mainly carried *MEN1* and *CDC73* pathogenic variants, while, notably, 26.9% of the second group harbored pathogenic variants such as *CDC73* and *CASR*. The “genetic” group was more frequently asymptomatic, while renal complications had the highest prevalence among other involvements within second group (71%) [26]. These two studies [26,27] pointed out the importance of genetic awareness in pediatric PHP despite negative family history, noting that 5–29% of the children that were initially considered sporadic cases actually harbored a pathogenic variant [26,27].

Of note, in dramatic cases like parathyroid (hypercalcemic) crisis, which is not responsive to standard medical therapy, or even off-label use of calcium-lowering agents such as denosumab [65], emergency parathyroidectomy should not be postponed in order to find out the results of the genetic analysis [66] (Table 3).

### 3.3. Lab Findings

Serum calcium and PTH values represent useful diagnostic and follow-up tools; sometimes, PHP is unexpectedly identified due to a high calcium value at the moment when the patient is admitted for an apparently unrelated complaint, such as an infection [68]. Increased values of calcium require prompt medical intervention, including the decision of administering bisphosphonates and cinacalcet or, in refractory cases, emergency parathyroid tumor removal [66]. For example, Hayashi et al. [66] described the case of an 11-year-old male with parathyroid (hypercalcemic) crisis, which mostly manifested as severe digestive complaints in association with a total serum calcium of 14.3 mg/dL, as well as a PTH value of 405 pg/mL. Despite standard care (intravenous fluids, furosemide, and calcitonin), the hypercalcemia rose to 16.5 mg/dL; pamidronate administration induced a drop to 14 mg/dL, which was followed by emergency parathyroidectomy that achieved control of the PTH excess [66].

A reduced, yet higher-than-normal value of PTH may be found after surgery, even after successful parathyroidectomies (for up to a 1-year duration), most probably in relation to a vitamin D deficiency or to a certain latency of PTH correction, which is currently less understood, as pointed out in the case of a 11-year-old patient with an ectopic parathyroid tumor resection (if genetic PHP is excluded) [68]. 

On the contrary, postoperative hypocalcemia associated with normal (or high, but lower than the preoperative level) PTH, as described in hungry bone syndrome, was reported in a few cases/series [15,36,69,70]. For instance, a 14-year-old male who was admitted for digestive complaints and lethargy was detected to have a PTH excess-associated brown pelvic tumor, with remission following parathyroidectomy, accompanied by transitory starvation bone syndrome [71]. Sharanappa et al. [28] reported on a cohort of 35 pediatric subjects with PHP, showing a prevalence of hungry bone syndrome of 34% [28]. Boro et al. [48] showed, in a series of 10 consecutive patients with PHP (a mean age of 16.7), a prevalence of 40% [48]. 

As expected from the data on adults diagnosed with a parathyroid carcinoma, this exceptional histological entity (that is described in less than 1% of pediatric cases with PHP) is associated with very high PTH values [72,73]. For instance, Rahimi et al. [73] reported on a 15-year-old female who initially presented with a PTH level of 2876 pg/mL (the immunohistochemistry report showed a Ki67 of 2–3%, and positive CK and CD31 reactions) [73]. The same authors [73] performed an analysis of published cases (from 1972 to 2020) and found 17 reports of parathyroid carcinoma in the pediatric population; total serum calcium varied between 12 and 20.7 mg/dL, and PTH levels were between 300 and 8638 pg/mL (the ages reported were between 8 and 16 years; the male to female ratio was 0.6; 60% of these children had palpable neck tumors; 50% of them displayed various bone complications; one out of five subjects had distant metastases, particularly pulmonary and lymphatic) [73]. Of note, a similar review from 2020 identified 12 published cases of parathyroid carcinomas in children, in addition to a new report of a 13-year-old male (with a baseline total serum calcium of 15.43 mg/dL and PTH of 980 pg/mL) [72]. 

Collaterally, we mention the assessment of vitamin D level status; some authors suggested that hypovitaminosis D is prone to be associated with postoperative hypocalcemia or hungry bone syndrome, or it impairs the remission of brown tumors while the correction of serum 25-hydroxyvitamin D through vitamin D replacement remits the component of secondary hyperparathyroidism that may be additional to PHP [74]. 

### 3.4. Imaging Tools 

Cervical ultrasound is the basic imaging assessment used, as seen in the adult population. A recent study concerning the present topic focused on preoperative imaging techniques, namely, cervical ultrasound and 99mTc Sestamibi scintigraphy. This was a single-center experiment in 32 subjects, 18 years old or younger (between 2003 and 2021), with an average age of 14.7 years, that were diagnosed with single parathyroid tumors (no ectopic adenomas were included; the median diameter of the adenomas was 2.85 cm). He et al. [45] found that echography had a sensitivity of 100%, while 99mTc Sestamibi scintigraphy was concordant in 93% of cases [45]. 

Ectopic (extra-cervical) parathyroid tumors are less likely to be detected through neck echography; thus, functional imaging to address the parathyroid glands is useful, with the most practical imagining technique being a 99mTc Sestamibi scan. For example, Sahu et al. [11] reported, in 2023, a 12-year-old female who was admitted for severe bone anomalies (limb deformities in association with a history of multiple fractures) and kidney stones. She had an intra-thymic parathyroid tumor, confirmed though Sestamibi scintigraphy, that required a thoracoscopic left thymectomy (using a Sestamibi-guided gamma camera during the procedure in order to locate the ectopic adenoma), with a good postoperative outcome [11]. On the other hand, Lenschow et al. [75] showed, in a large study of 628 patients with PHP (only a few females were younger than 18, and most of the studied population included adults, with an overall median age of 58), that, in cases with negative cervical ultrasound results, 99mTc Sestamibi identified the parathyroid tumors in only 25.4% of them, while ^11^C-methionine or ^11^C-choline PET/CT (positron emission tomography/CT) identified the lesions in 79% of this subgroup [75]. One case of a teenager with PHP showed a positive Lincoln sign (“black beard sign”), due to mandibular uptake of the tracer during 8 F-fluorocholine PET/CT [76]. 

Another report of an ectopic parathyroid adenoma showed the utility of 99mTc Sestamibi (SPECT/CT) in association with 4-dimensional CT [68]. Similarly, SPECT/CT was used to identify an intra-thymic adenoma in a 15-year-old male [10]. Moreover, a study on 23 children and teenagers (4/23 with germline pathogenic variants: 3/4 with *CDC73* and 1/4 with *CASR*) with PHP showed that dual-phase CT provided good sensitivity (91.3%) and specificity (99.5%) in single-gland disease; 95% of the patients with single adenoma showed post-operative remission within the first 18 months after the initial use of CT data [46].

Another aspect of imaging assessment in PHP includes the diagnosis of brown tumors, either as incidental findings, or due to local bone pain, deformities, or walking dysfunctionality. They are expected to remit after successful parathyroid tumor removal; thus, serial scans, for instance, with X-ray, CT or magnetic resonance imaging, are mandatory [38,74,77]. Omi et al. [78] reported an impressive case of parathyroid carcinoma, operated on in a patient at the age of 13, who then survived without recurrence for more than 40 years, due to multimodal management in terms of diagnosis and therapy; notably, the current recurrence was identified through ^11^C-methionine-positive features of the tumor (i.e., 99mTc-methoxy-isobutyl-isonitrile- and ^18^F-fluorodeoxyglucose-negative) [78].

Overall, imaging analysis is mandatory in order to choose the type of surgical approach; however, high-volume surgical teams are expected to provide the highest success rates, due to intraoperative localization and the use of intraoperative PTH assays. As mentioned, the importance of evaluating the genetic background of a child with PHP is reflected in the methods of addressing the sites of the tumors (such as multi-glandular disease of ectopic lesions in genetic PHP) and the type of parathyroidectomy. 

### 3.5. Surgical Procedures 

Parathyroidectomy represents the single curative option for PHP, regardless of whether it occurs in the pediatric or adult populations, with a general cure rate of 95–99%, the exceptions being persistent PHP due to inefficient location of the tumor before and during surgery, multi-glandular parathyroid disease (typically present in genetic PHP), and synchronous ectopic adenomas [79].

Thoracoscopic parathyroidectomies, including those conducted through minimally invasive techniques, such as video-assisted thoracoscopic surgery (VATS), that are facilitated by having a precise preoperative location of the adenoma, have been safely used in selected adult cases of ectopic or extremely large orthotopic parathyroid tumors; the similar pediatric application also seems safe, according to current knowledge, despite a low level of statistical evidence [11,80,81,82]. For example, Vitale et al. [69] reported, in 2022, the first pediatric case of PHP-associated ectopic (intra-thymic) mediastinal adenoma complicated by bilateral slipped capital femoral epiphysis (the same team analyzed similar, previously published, cases and identified 13 reports with unilateral epiphysis lesions). The parathyroid tumor (which was confirmed, preoperative, with 99Tc-Sestamibi and CT scans) was successfully removed via a thoracoscopic resection. The 12-year-old male developed post-surgery hungry bone syndrome and vitamin D deficiency–related secondary hyperparathyroidism [69]. Another case of bilateral slipped capital femoral epiphysis was reported in a 15-year-old male, who also suffered from other complications of PHP (such as polydipsia, polyuria, and weight loss); however, the patient first underwent orthopedic correction (due to pathological report confirming brown tumors) and, further on, parathyroidectomy of the single adenoma was performed [77]. Moreover, another unilateral case was reported, in relation to an ectopic adenoma, by Badhe et al. [83]; Seo et al. [10] added a case of a teenager who underwent VATS–based removal of an intra-thymic adenoma [10]. Alternatively, an ectopic site within the piriform sinus of a 9-year-old child required a reperformed parathyroidectomy [84]. Overwhelmingly, most recent reports include thoracoscopic resection of ectopic adenomas [11,29,68,69]. Additionally, a robot-assisted surgical procedure for an ectopic (thymic) parathyroid adenoma resection was successful, not only to control the biochemical findings, but also, to control the associated psychiatric manifestations in the affected teenager [85].

In 2021, Flokas et al. [68] published a review of ectopic parathyroid adenomas in patients 18 years and younger and identified 33 cases (published between 1982 and 2019). Prior reports suggested that ectopic forms are identified in 5–26% of the children confirmed with PHP. In this analysis, there was relatively equal sex distribution; 2 out of the 33 tumors were carcinomas; half of the ectopic adenomas were located at the thymus level and 17% of all tumors were unexpectedly detected. Other comorbidities included bone and kidney complications in addition to a MEN1-associated panel (1/33) [68]. 

The high frequency of mediastinal (intra-thymic) site occurrence among ectopic tumors suggests that the thoracoscopic approach and minimally invasive techniques are stepping stones in modern parathyroid pediatric surgery, which is otherwise challenging, not only in relation to the adenoma site, but also with the patients’ ages and preoperative complications. For example, the analysis by Sharanappa et al. [28], performed between 1989 and 2019, pointed out that, among 35 pediatric subjects with PHP (8.5% of whom were diagnosed with familial PHP), 91% had a parathyroid adenoma; minimally invasive parathyroidectomies were performed in 40% of all cases, and the cure rate was 97%, which was consistent with prior reported data [28]. One retrospective study from 2022 analyzed a single (high-volume) surgeon’s experience from 2002 to 2020 and showed a safe profile of the parathyroidectomy in 19 pediatric outpatients (with a mean age of 14.1). The surgical procedure varied, from 8/19 unilateral parathyroidectomies to 9/19 trans-cervical thymectomies. Preoperative findings showed that fatigue was the most frequent clinical element (affecting 62% of the subjects); lab assays showed a mean total serum calcium of 11.7 mg/dL and an average PTH of 102.3 pg/mL. The cohort included two genetic PHPs (MEN1 and MEN2). A postoperative calcium correction was identified (a mean value of 9.6 mg/dL was found at 6 months following the procedure; 1/19 subjects experienced transitory hypocalcemia), as well as normalization of elevated PTH (a mean of 29 pg/mL at 6 months after surgery; 1/19 of the subjects was confirmed with permanent hypoparathyroidism) [47].

### 3.6. Outcome 

While parathyroidectomy is recognized worldwide to provide the cure for hyperparathyroidism in most cases, the management of PHP-associated hypercalcemia represents a heterogeneous domain. Calcimimetics are approved for selected cases of adult primary PHP (not only renal hyperparathyroidism), depending on country protocols; their off-label use in infants and teenagers with PHP seems to be a new area that remains a matter of debate. The drug may induce severe hypocalcemia, as well as cardiac rhythm anomalies such as prolonged QT interval; however, some data showed its efficacy, being seemingly safe when offered to young patients for a limited period of time [86,87]. A multicenter French study on 18 patients (with a median age of 10.2; range between 2 and 14.4 years) included 13 individuals with primary PHP (10 of them with known genetic PHP-carrying pathogenic variants with respect to the *CASR*, *CDC73*, and *MEN1* genes). Overall, under a progressively increased dose over a median of 2.2 years, cinacalcet decreased PTH and serum calcium levels, with stable alkaline phosphatase and no significant side effects [18].

The overall outcome in PHP has been analyzed in a few studies, according to our 3-year analysis, as previously mentioned. Additionally, another retrospective, tri-center study from 2020 outlined the outcomes after parathyroidectomy on 86 patients, with a median age of 17 (64% were females); the 6-month cure rate was 98%. Of note, 25% of the tumors were ectopic (and half of them were thymic), while preoperative identification of adenoma was conducted with 99mTc Sestamibi in 58% of cases, and with neck ultrasound in 43% of the patients [5].

We identified a singular study of its kind, a case–control analysis in pediatric vs. adult subjects, who were referred to surgery at a single (high-volume) center of surgery (between 2004 and 2017). The juvenile group (N = 14) demonstrated a more severe clinical picture vs. the adult group (N = 28); the most frequent complication was bone disease (affecting 42% of the subgroup), while the adults were mostly asymptomatic (39%) [51]. Moreover, the authors published a second (retrospective) study comparing children (≤15 years, N = 6, male-to-female ratio of 2) with teenagers (>15 and ≤20 years, N = 8, male-to-female ratio of 1 to 1.7) who had renal stones in 0% vs. 62% of subjects, and statistically significant higher calcium and PTH levels in the younger vs. older group [52].

A mostly unexpected resolution of PHP was detected after a bilateral adrenalectomy for Von Hippel–Lindau Disease-related bilateral pheochromocytoma with ectopic (intra-adrenal) PTH expression; this was the case of a 17-year-old teenager admitted for pheochromocytoma-related diabetes mellitus and high blood pressure. This represents the only patient harboring the VHL pathogenic variant that we could identify within the mentioned timeframe of our research [67] (Table 4).

## 4. Discussion 

PHP in children and teenagers is associated with a heterogeneous spectrum, from baseline picture to long-term follow-up. Our 42-month sample-based study was related to several major points, as follows. 

### 4.1. Changes in Terminology over the Years concerning Parathyroid Tumours 

The last 3 years brought forth new terminologies, such as those included in the WHO 2022 classification; for example, “multi-glandular” is now used instead of the prior “hyperplasia”, there are new definitions for carcinomas and atypical tumors (previously described as “adenomas”), including the distinct subgroup of “parafibromin-deficient neoplasms”, in association with the increasing importance of genetics, immunohistochemistry, and molecular biology assessments in parathyroid tumors [88]. Nevertheless, gathering data from different publications might help to share common ground, in order to achieve a certain level of understanding and knowledge. 

We restricted our research to PHP in order to avoid conditions with different pathogenic traits; that is why we did not include children diagnosed with acute or chronic kidney damage-related hyperparathyroidism, involving the parathyroid glands as a secondary event following kidney lesions, regardless of their type (including post-transplant PTH status); for the same reason, we also did not include Bartter syndrome, which can be associated with nephrocalcinosis and increased PTH values [89,90,91,92]. We also did not include studies that specifically addressed hypovitaminosis D-associated secondary PTH value increase; however, the importance of vitamin D supplementation to correct its deficiency was also found in the pediatric population co-diagnosed with PHP, as similarly seen in adults with the same conditions [93]. 

PHP remains a challenging topic that concerns a multidisciplinary perspective. The index of clinical suspicion is far from being generous, but awareness is essential in order to avoid a hypercalcemia-associated life-threatening event or progressive skeletal and renal anomalies due to long-term uncorrected PTH and high calcium values. 

### 4.2. Neonatal Primary Hyperparathyroidism 

Alternatively, we conducted a second analysis with respect to severe neonatal hyperparathyroidism. We applied the same research strategy and utilized an additional term, “neonatal”; thus, we identified 49 papers and selected 9 of them (including original case reports and series, for a total of 15 patients). This represents a dramatic condition diagnosed within the first six months after birth, usually in relation to a genetic background involving *CASR*. Familial hypocalciuric hypercalcemia of types 1 to 3 underline *CASR* anomalies that induce a mild elevation in serum calcium and PTH, starting from childhood and continuing into adulthood (which is distinct from PHP, with parathyroidectomy being unnecessary) [94,95,96]. Associated pathogenic variants involve the *CASR* gene on chromosome 3q 13.3–21 (loss of mutation), with more than 300 mutations described so far (60% of all familial cases), but also involving the *GNA11* (guanine nucleotide-binding protein subunit alpha-11) gene, as well as the *AP2S1* (adaptor protein complex 2 subunit sigma) gene, affecting 5% and 20%, respectively, of all individuals confirmed with familial hypocalciuric hypercalcemia [97,98,99]. However, the inheritance of the condition (homozygote status) may lead to dramatic consequences in terms of neonatal presentation, with a high morbidity (due to metabolic, bone, nutritional, and neurodevelopmental consequences) and mortality [100]. 

Extremely rare heterozygous mutations may manifest with a dramatic depiction of hypercalcemia at birth (these cases are prone to be transitory, as opposed to homozygous) [101]. For example, a new case was reported in 2022, in a boy who inherited the mutation from his father, who was an asymptomatic heterozygous carrier of the same gene. His admission was based on a severe clinical picture of hypercalcemia (with lethargy and bradycardia) that was particularly managed with pamidronate and cinacalcet, with poor response; thus, parathyroidectomy was mandatory. Of note, the subject had a sibling who died of the same condition at the same age, which suggests dramatic consequences for this genetic type of PHP at the newborn stage [102]. 

Another dramatic case was reported in a newborn with abnormally high calcium and PTH values (of 12 mg/dL and 1120 ng/dL, respectively); he was treated for 13 months with cinacalcet and a parathyroidectomy was finally performed at the age of 17 months. Calcimimetic drugs delayed the need for surgery, thus representing a valuable treatment solution, even for cases with a modest response to this medication [103]. Another 7-month-old girl, who underwent parathyroidectomy for the same condition, was reported [104]. The first such report from Sudan included three siblings carrying a pathogenic homozygous missense variant of *CASR* and, since medical alternatives were not available, a parathyroidectomy was performed [105]. A 7-month-old male infant received a parathyroidectomy twice, since the (thymus) ectopic parathyroid tissue could not be located before surgery, and the decision to conduct a total thymectomy was made following persistent PTH values after the initial removal of all four parathyroid glands [106]. The use of an intraoperative PTH assay may serve to confirm the success of the surgical procedure [107]. Of note, the youngest patient receiving cinacalcet for homozygote status of the *CASR* gene—c.1836G>A (p.G613E)—was a 2-day-old newborn (in a report from 2020). The child was started on the medication due to very high calcium and PTH levels (14.4 mg/dL and 1493 pg/mL, respectively) during her first day of life; this case introduced the longest period of medical therapy with cinacalcet to avoid parathyroidectomy (18 months) [108].

Awareness of this particular type of PTH excess-related condition is essential, and it should be taken into consideration when addressing the larger frame of infancy-associated parathyroid neuroendocrine neoplasia (Table 5).

### 4.3. Calcium-Lowering Drugs in Children with PHP

As previously mentioned, cinacalcet was studied in one study (N = 18 patients with a median age of 10.2 years) [18], and for the case of one 14-year-old teenager [36]. The duration of therapy varied, from 3 months to a median of 2.2 years [18,36]. It has been suggested that its preoperative use may increase the risk of post-parathyroidectomy hungry bone syndrome that requires prompt calcium and calcitriol intervention, but the level of pediatric evidence remains low [36].

We identified two single case reports of denosumab administration in order to control unexpectedly high hypercalcemia [65]. One teenager was diagnosed with a parathyroid adenoma; he was first admitted for fractures and digestive complaints. The subject’s very high values of total calcium (16.71 mg/dL) and PTH (2151 pg/mL) required prompt intervention; 60 mg of denosumab was administered (in addition to standard care), and his hypercalcemia rapidly reduced within 12 h. Mamoedova et al. [65] considered that controlling hypercalcemia via medical therapy allowed for a useful timeframe in which to obtain the results of the genetic analysis (in this case, it revealed a heterozygote status of the *CDC73* gene). A genetic condition might require a multi-glandular resection and/or a distinct surgical approach. The 16-year-old patient underwent parathyroidectomy on day 22 after denosumab injection, with tumor removal being followed by hypocalcemia. A potential role of the drug in inducing prolonged low serum calcium values might therefore be incriminated in subjects diagnosed with PHP and renal hyperparathyroidism [109,110]. However, the adequate administration of calcium and alfacalcidol controlled the tetany in this case [65]. 

The second case of juvenile denosumab use was in a 13-year-old boy diagnosed with a PHP-associated parathyroid carcinoma, complicated with brown tumors and post-parathyroidectomy hungry bone syndrome. In this situation, after standard care of preoperative hypercalcemia with intravenous fluids, two doses of subcutaneous calcitonin (4 U/kg) were followed by two consecutive doses of pamidronate (0.5/kg) that were not efficient; thus, 60 mg subcutaneous denosumab was offered to the young patient (on the fourth day since first admission), resulting in a normal calcemic value after another 4 days [72]. Generally, off-label use of calcium-lowering agents that are only approved for adults (including denosumab) requires distinct ethical precautions, such as Local Committee of Ethics approval and informed consent from the patients (depending on their age) and their parents. 

### 4.4. Paediatric PHP: Osteoporosis and Fracture Issues

Neuroendocrine tumors of the parathyroid gland might be associated with bone involvement, such as fragility fractures, reduced bone mineral density (BMD) (implicitly, of Z-scores in children), deterioration of bone microarchitecture, and even impairment of peak bone mass achievement during the teenage years. However, the decision of therapy against osteoporosis is controversial, since successful parathyroidectomy might restore the normal bone–skeletal dynamics, depending on the patient’s age, sex, and pubertal stage [2,111,112]. Fukaya et al. [35] presented the case of a 12-year-old girl with non-syndromic PHP who was confirmed with low BMD according to a Z-score of −2.7 SD (on admission, total serum calcium was 13.7 mg/dL and PTH level was 321 pg/mL). After successful parathyroidectomy of an adenoma, of 2.5 cm at its largest diameter, the PTH level decreased to a value of 10 pg/mL without associated symptomatic hypocalcemia. Good clinical and lab parameter evolutions were associated with an increase in BMD, according to a lumbar Z-score of −1.9 SD 6 months after parathyroid removal, with a further measurement of −1.2 SD recorded after another 10 months [35]. 

The correct timing of performing and repeating DXA (Dual-Energy X-ray Absorptiometry) in these young patients is still a matter of debate, unless a clinical fracture or osteoporosis diagnosis is difficult to confirm. Moreover, the interpretation of DXA assessments should take into consideration physiological changes during growth and puberty, including the achievement of peak bone mass [113]. A recovery of prior weight loss due to hypercalcemia-associated appetite loss, recurrent abdominal pain, and even depressive mood might be correlated with an increase in DXA-BMD. However, the decision of using bisphosphonates should be cautiously considered in cases with fractures, since parathyroid tumor removal will most likely improve the bone status in a matter of months [114,115,116]. On the other hand, very low Z-scores were reported in patients with multiple fractures and those with brown tumors; for instance, de Silva et al. [40] reported on a 17-year-old male teenager with a Z-score of −5.8 who had a severe presentation, with multiple fractures that required the use of a wheelchair [40].

As an alternative to a DXA scan (which turned out to be normal), Lenherr-Taube et al. [72] assessed microarchitecture and strength via high-resolution peripheral quantitative CT (HRpQCT) at the levels of the distal tibia and distal radius in a 13-year-old patient with PHP; the scans, performed 1.5 years after successful parathyroid surgery, showed a good outcome in terms of osteolytic lesion resolution, an increased volumetric density, an increased number of trabeculae, as well as improved cortical thickness and density (within healthy control ranges) [72]. HRpQCT might represent an alternative for skeletal assessment in pediatric PHP; there are a few similar studies in adults with PHP, as well. However, currently, the level of overall statistical significance remains low; also, routine use of HRpQCT is limited by access to the device. Moreover, normal (healthy) reference population parameters (normative data) according to biological age, sex, pubertal stage and geographic area must be provided [117,118]. Lately, the lumbar DXA-based trabecular bone score (TBS), as a practical tool for bone microarchitecture assessment, was applied in individuals younger than 18 years old to a lesser extent than in adults, but the topic continues to expand, and we anticipate valuable pediatric studies in PHP, as well [119,120]. As seen in other malignancies of both children and adults, the overall bone status should be regarded in relation to the primary tumor (such as a parathyroid carcinoma) and in relation to the changes resulting from surgery. 

### 4.5. Trans-Pandemic Insights into Paediatric PHP

Our analysis covered the COVID-19 pandemic era; during this period of time, many medical and surgical conditions were reshaped or associated with new entities, due to coronavirus infection or to new COVID-related regulations [121,122,123]. With respect to PHP, a delay in hospital admission and, potentially, a more frequent indication of using calcium-lowering agents, such as cinacalcet or denosumab to function as a bridge to the surgical procedure (if it was not considered an emergency), were registered amid the first waves [124,125]. However, we found no pediatric studies on this specific matter. Moreover, compressive neck symptoms should prompt consideration of not only thyroid and parathyroid tumors, but also infection-associated clinical pictures [126,127,128]. 

Overall, after reviewing almost 350 papers, we included the following in our final analysis: 48 original studies with different levels of statistical significance (which included 14 studies of at least 10 subjects per study and 34 case reports and series of fewer than 10 patients per study) and a collateral checkup of neonatal cases (thus identifying another 9 case reports and series), which is a distinct subgroup that we previously mentioned [99,101,102,103,104,105,106,107,108].

The main study-sample-based analysis showed that, except for a case–control study, all of the studies were retrospective with multicenter (N = 5) or single-center (some including a single-surgeon experience) studies (N = 7). Sample size data identified small cohorts of 10 to 19 patients (10, with cohorts of 14, 18, and 19 subjects per study); medium-sized studies, including between 23 and 36 individuals (with cohorts of 23, 32, 35, and 36 subjects per study); and large studies, enrolling between 63 and 83 children and teenagers (with cohorts of 63, 66, 80, and 83 subjects per study); thus, these studies included a total of 493 individuals with PHP. Some case reports included articles with specific genetic testing: five reports with one case, and two reports with two cases (N = 7, N = 9). Non-genetic information was included in 27 papers (23 with a single patient per report, 3 with 2 patients per series, and 1 with 3 subjects per series; N = 27, N = 32). Thus, a total of 41 patients from isolated reports were included (N = 534 total subjects with PHP in childhood and teenage years, according to our methods). 

The admissions of the patients within the mentioned study designs occurred between 1989 and 2021 (different timeframe combinations). The mean ages of the patients were as follows: 10.2 (the lowest mean value) in some studies, and around 14 years (for example, 14, 14.1, 14.7, and 15.2 years) in other studies and case reports, as well as around the age of 17 (for instance, 16.7, 17, and 17.3) in case series [5,18,26,27,28,45,46,47,48,49,50,51,52]. We exceptionally included four studies that applied a (maximum) cutoff age of 20–21 (rather than 18), but for which the average and median ages of the entire studied populations were below 18 [5,26,27,52]. Case reports included patients within an age range of 9 to 17 years [9,10,11,15,29,34,35,36,37,38,39,40,44,66,68,69,70,71,72,73,74,77,78,83,84,85]. Overall, no clear female predominance was registered, as is seen in the adult population with PHP [51,52]. 

Most studies were descriptive. A single study compared adults (N = 28) with children (N = 14) [51], while another showcased infants vs. teenagers [51]. Genetic PHP and apparently sporadic PHP were compared in four studies [18,26,27,50]. Concerning the assessments within the studies, we identified four major types of endpoints, as follows: particular imaging data, such as ultrasound, 99mTc Sestamibi scintigraphy, and dual-phase computed tomography [45,46]; a second cluster included meticulous gene testing, or patients with familial syndromes such as MEN1, etc. [18,26,27,46,47,49]; a third point related to preoperative findings in terms of clinical presentation [5,26,27,28,48,51,52]; meanwhile, the fourth type of evaluation provided surgical outcomes, postoperative issues, and an alternative to surgical approach in the form of a study on preoperative cinacalcet exposure with a 2.2-year median duration [18].

Analyzing the type of gland involvement showed involvement of a single gland in 85% of sporadic cases, and in 19% in genetic PHP cases [27]; localization studies found 100% sensitivity for neck ultrasound in cases with single, non-ectopic tumors, with a 98% concordance with 99mTc Sestamibi [45]. A 91% sensitivity for dual-phase CT for single-gland tumors was revealed in one study [46], in which 25% of the lesions were ectopic parathyroid tumors (mostly intra-thymus, representing one-half of the ectopic presentations) [5]. Case reports included another 9 out of the 41 patients with ectopic parathyroid adenomas [10,11,29,34,68,69,83,84,85], and 3 parathyroid carcinomas [72,73,78]; moreover, 8 subjects (N = 8/41) had brown tumors [15,36,38,40,71,72,74,77].

Genetic PHP mostly involved the *MEN1* gene, followed by *CDC73*, *CASR*, *RET*, and *CDKN1B*; a maximum prevalence of 32–52% with respect to genetic PHP was reported, and an additional subgroup representing 5–26.9% of the patients was reclassified from apparently sporadic to genetic PHP [26,27,50]. Gene assays from isolated reports included *MEN1* pathogenic variants in four papers [53,61,62,64], *CDC73* variants in two articles [17,65], and a *VHL* mutation in one subject [67]. Novel findings in the genetic field included a germline *MEN1* mutation heterozygous variant c.105_107dupGCT [62] and a new pediatric phenotype with ectopic GHRH pancreatic overproduction in MEN1 [61].

Symptomatic PHP was reported in 71% of all cases (with a mean age of 17.3) [27], while asymptomatic PHP was found in 60% of genetic PHP cases, and in 0% for sporadic PHP cases [26]. Renal involvement (including kidney stones) was recorded to affect the following: 10.5% of a cohort with genetic PHP, 71% of the cases diagnosed with sporadic PHP [26], 50% of a cohort with a mean age of 16.7 [48], and 29% of another cohort with a mean age of 15.2 [28], from 0% (in infancy) to 50–62% (teenagers) [52]. Bone anomalies affected 50% of a cohort with deformities, while 30% of the patients had fractures [48]; 83% of the children were affected by some type of skeletal involvement [28], while 62% of the studied population was affected in another study [52]. Gastrointestinal issues affected 40% of one cohort [48], but the data in this domain are heterogeneous. 

The cure rate through parathyroidectomy was 97% [28] to 98% [5]. Recurrent PHP was reported in 2% of sporadic PHP cases, and in 38% of familial PHP cases [27]. Hungry bone syndrome was found with a maximum rate in 34% of the cases in one study [28], and in 40% of another [48]. Case reports identified another seven subjects with the same post-parathyroidectomy hypocalcemia-based syndrome; a potential connection with ectopic presentation or brown tumors was suggested, but there are limited data from a strictly statistical perspective [15,36,69,70,71,72,74]. Minimally invasive thoracoscopic approaches seem safe, and they should be used for ectopic tumors when possible [11,80,81,82].

We are aware of the limitations of a narrative review mainly focused on PubMed–based research for sample-based analysis, but this allowed us to have a more flexible approach than a systematic review/meta-analysis would allow, due to the heterogeneity of the studied parameters. Within the timeframe of our research, we identified a few reviews with different objectives in pediatric PHP, as mentioned. For instance, Bin Yahib et al. [29] analyzed 13 previously published papers (N = 331 children, teenagers and young adults) studying recurrent abdominal pain as feature of PHP [29]. Lee et al. [39] studied prior reports of young patients with genu valgum and PHP and identified 23 such cases [39]. Rahimi et al. [73] analyzed 17 previously published studies with regard to pediatric parathyroid carcinoma [73], while another review identified 12 individuals confirmed with the same histological diagnosis and added 1 new report (N = 13) [72]. In 2021, Flokas et al. [68] published a review of 33 ectopic parathyroid juvenile adenomas [68]. 

Due the level of statistical evidence concerning the area of pediatric PHP publications, our study- and case-sample-based analysis, which included 48 articles and a total of 534 published cases of PHP, represents one of the largest of its kind.

## 5. Conclusions 

PHP in children and teenagers represents a challenging and dynamic field with a necessarily multidisciplinary approach. Recent data suggest that infants and patients with an already-known genetic/familial condition are prone to asymptomatic PHP. An infant with PHP may be the index case in their family for genetic conditions such MEN1. The panel of bone, renal, and digestive complications varied, and an index of suspicion should be kept in mind. One out of four patients had ectopic parathyroid tumors, most probably mediastinal. The importance of preoperative imaging assessment, such as 99mTc Sestamibi scintigraphy, was closely related to the usefulness of genetic testing before parathyroidectomy in providing adequate information that influenced the surgical technique. Thoracoscopic procedures showed good results, as seen in adults nowadays, but the level of evidence remains low in this particular instance. Awareness of pediatric PHP is the key factor to benefiting our young patients. 

## Figures and Tables

**Figure 1 biomedicines-11-02810-f001:**
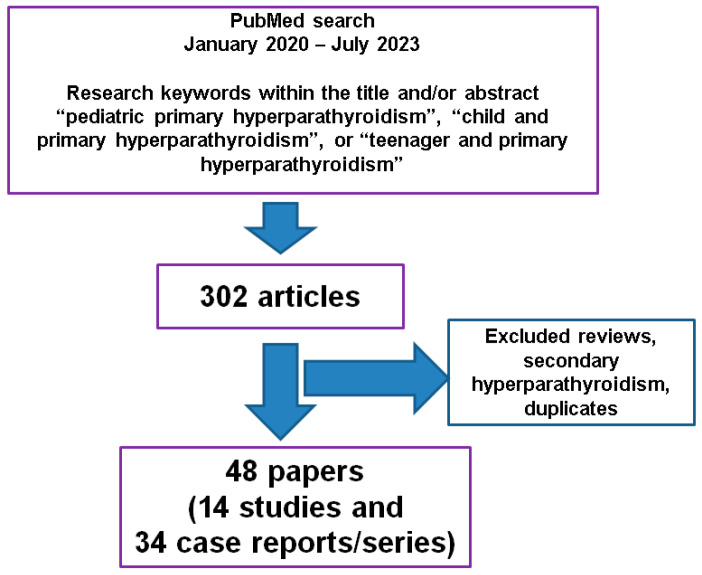
The workflow of research according to our methods.

**Table 1 biomedicines-11-02810-t001:** The search criteria according to our methods.

Inclusion Criteria	Exclusion Criteria
Studied population: children and teenagers (patients 18 years old or younger)	Reviews, editorials
PubMed access of the papers	Experimental studies
Full-length, English-language articles	Secondary (vitamin D-associated) hyperparathyroidism and/or rickets
Research keywords within the title and/or abstract: “pediatric primary hyperparathyroidism”, “child and primary hyperparathyroidism”, or “teenager and primary hyperparathyroidism”	Secondary or tertiary (renal) hyperparathyroidism
Original studies (studies, case series, case reports) of any design (regardless the level of statistical evidence)	Primary hyperparathyroidism in pregnancy
Time frame (by publication date): January 2020–July 2023	Familial hypocalciuric hypercalcemia Bartter syndrome Inherited conditions of mineral metabolism
	Mixed (adult and pediatric) studies, unless a specific analysis of the pediatric data was provided

**Table 2 biomedicines-11-02810-t002:** Studies addressing PHP in children and teenagers (with different types of outcomes, and at least 10 patients per study) according to our methods; the list starts with the most recent publication date [5,18,26,27,28,45,46,47,48,49,50,51,52].

First AuthorYear of PublicationReference Number	Study DesignStudied Population	Outcome
He2023[45]	Single-center study (between 2003 and 2021)N = 32 patients with PHPAge: ≤18 years old (mean age of 14.7 ± 2.5 years)Parathyroid tumor size: 1–5.8 cm (mean size of 2.85 cm)	Cervical ultrasound: 100% sensitivity for single parathyroid tumors (none ectopic)99mTc Sestamibi scintigraphy: concordant with ultrasound results in 98% of cases (N1 = 30)N2 = 2 patients with multi-glandular parathyroid disease according to 99mTc Sestamibi, but not with ultrasound assessment
Sharma2023[46]	Single-center imaging studyN = 23 children and teenagers with PHP (4/23 with germline mutations: 3 with *CDC73* and 1 with *CASR*)	Dual-phase computed tomography provided good sensitivity (91.3%) and specificity (99.5%) in single-gland disease
Szabo Yamashita2022[27]	Single-center retrospective study (between 1994 and 2020)N = 66 with PHP (61% females)Age: ≤21 years old (mean age of 17.3 years)	71% symptomatic PHP32% known with genetic PHP (mostly *MEN1*)5% of apparently sporadic cases were genetic PHPSporadic vs. familial PHP: 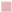 Single-gland disease: 85% vs. 19% (*p* < 0.00001) 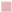 recurrent PHP: 2% vs. 38% (*p* = 0.0004) 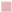 postoperative recurrence time: 61 vs. 124 months (*p* = 0.001)
Sharma2022[26]	Single-center retrospective study (between January 2020 and January 2021)N = 36 patients with PHP (55% males)Age < 20 years (median age of 17 years)N1 = 10 genetic/familial PHPN2 = 16 apparently sporadic PHP	N1: 90% with pathogenic variants: *MEN1* gene (8/10), *CDC73* (1/10)N2 = 26.9% with pathogenic variants/homozygotes: *CDC73* (4/26), *CASR* (3/26)N1 vs. N2: 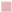 asymptomatic PHP: 60% vs. 0% (*p* < 0.001) 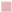 lower baseline PTH: 237 vs. 1369 pg/mL (*p* = 0.001) 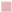 lower maximum tumor diameter (0.9 vs. 2.2 cm, *p* = 0.01) 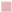 renal complications: 10.5% vs. 71.4% (*p* = 0.01)
Bernardor2022[18]	Multicenter study N = 18 patients with PHP receiving cinacalcetMedian age of 10.2 years (N1 = 10 genetic PHP involving *CASR*, *CDC73*, and *MEN1* genes)	Median duration of therapy: 2.2 yearsDose: 0.7 (0.6–1) mg/kg/day → 1 (0.9–1.4) mg/kg/dayPTH drop (*p* = 0.01)Hypercalcemia drop (*p* = 0.002)Nephrolithiasis: 1/13
Ramonell2022[47]	Single-center (single surgeon) studyN = 19 patients who underwent parathyroidectomy (as outpatients) for PHPMean age of 14.1 years N1 = 1 case with MEN1N2 = 1 case with MEN2	8/19 patients had unilateral parathyroidectomies 9/19 patients had trans-cervical thymectomies1/19 subject had transitory hypocalcemia1/19 subject had permanent hypoparathyroidism
Boro2022[48]	Single-center studyN = 10 patients who underwent parathyroidectomy Mean age of 16.7 years	90% (of the patients) had muscle and skeletal complaints 50%: bone deformities 50%: kidney stones30%: fractures 40%: gastrointestinal complaints30%: pancreatitis40%: hungry bone syndrome
Shariq2021[49]	Retrospective study N = 80 patients with MEN1Age: ≤18 years old (median age of 14 years)	80% of the patients had PHP (70% of them underwent a parathyroidectomy)
El Allali2021[50]	Retrospective studyN = 63 patients with PHP who underwent genetic analysis	52% had genetic PHPYounger group (94%) had *CASR* mutationsOlder group presented other mutations (*MEN1*, *CDC73*, *RET*, and *CDKN1B*)
Sharanappa2020[28]	Retrospective study (between 1989 and 2019)N = 35 patients with PHPMean age of 15.2 years	94% of the patients had symptomatic PHP83%: skeletal anomalies29%: renal complications8%: familial PHP97%: cure rate via parathyroidectomy2.8%: hypercalcemic crisis 34%: hungry bone syndrome
Rampp2020[5]	Triple-center retrospective study (between 1997 and 2017)N = 83 patients with PHP who underwent parathyroidectomy (64% females)Age: ≤21 years old (mean age of 17 years)	25% of the patients had ectopic adenomas (59% of them were intra-thymic)98% of the patients had a 6-month cure rate
Jovanovic2020[51]	Case–control, single-center (high-volume surgery) studyN1 = 14 patients with PHP (age ≤ 20 years)N2 = 28 adults with PHP	N1: high frequency of bone disease (42% of the patients from N1)N2: high frequency of asymptomatic presentation (39% of the patients from N2)Female to male ratio: 1 to 1 (N1); 8 to 1 (N2) (*p* = 0.005)
Zivaljevic2020[52]	Retrospective studyOut of 1363 patients, N = 14 patients with PHP (age < 20 years representing 1% of entire cohort)N1 = 6 children with PHP (aged ≤ 15 years)N2 = 8 teenagers with PHP (age: >15 and ≤20 years)	N1 vs. N2: 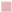 male to female ratio: 1 to 1 (N1); 1 to 1.7 (N2) 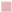 renal stones: 0% versus 62% 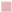 Most frequent complications: bone (62%) vs. renal (50%) 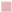 1 case of parathyroid carcinoma (N1)

Abbreviations: N = number of patients; PHP = primary hyperparathyroidism; PTH = parathyroid hormone; MEN = multiple endocrine neoplasia; Tc = Technetium.

**Table 3 biomedicines-11-02810-t003:** Case reports and series (<10 patients/article) with data concerning genetic profiling in pediatric PHP, according to our methods (the list starts with the most recent publication date) [17,53,61,62,64,65,67]; additional studies (≥10 patients/study) that also include genetic analysis are introduced in Table 2. Data were provided by Sharma [26,46], Szabo Yamashita [27], Bernardor [18], Ramonell [47], Shariq [49], El Allali [50], Sharanappa [28].

Pathogenic Variant	First Author Year of Publication Reference Number	Index Case
*MEN1*	Petriczko 2022 [64]	17-year-old male diagnosed with PHP and pancreatic neuroendocrine tumor Family data: father with PHP; sister with PHP, pancreatic neuroendocrine tumor, and central ganglioglioma
	Srirangam Nadhamuni 2021 [61]	At the age of 10: resection of an insulinoma At the age of 15: parathyroidectomy for PHP At the age of 18: gigantism due to ectopic GHRH production (pancreatic neuroendocrine tumor)
	Cho # 2021 [53]	12-year-old female was the daughter of the proband PHP, prolactinoma, pancreatic neuroendocrine tumor; Frame-shift mutation: NM_130799.1:c.1546dupC (p.Arg516Profs∗15)
	Stasiak 2020 [62]	16-year-old female diagnosed with PHP and pituitary microadenoma Novel *MEN1* germline pathogenic variant (heterozygous variant c.105_107dupGCT) Family data: father with the same pathogenic variant (PHP and pituitary neuroendocrine tumor)
*CDC73*	Blackburn 2022 [17]	14-year-old male with heterogeneous gene deletion (symptomatic hypercalcemia due to a single PT adenoma) 10-year-old female with known autosomal dominant mutation (symptomatic hypercalcemia and bilateral PT adenoma)
	Mamedova 2020 [65]	16-year-old female admitted for fractures, nausea, vomiting, weight loss (heterozygote status) 60 mg denosumab → parathyroidectomy → hypocalcemia
VHL	Belaid 2020 [67]	16-year-old female admitted for diabetes mellitus and high blood pressure due to pheochromocytoma Ectopic PTH (adrenal) production → remission of PHP after bilateral adrenalectomy

Abbreviations: GHRH = Growth-Hormone-Releasing Hormone; MEN = multiple endocrine neoplasia; VHL = von Hipple–Lindau; PT = parathyroid; PHP = primary hyperparathyroidism; # = in this case, the child was the daughter of the proband.

**Table 4 biomedicines-11-02810-t004:** Case reports and series (<10 patients per article) of pediatric PHP (the list starts with the most recent publications) [9,10,11,15,29,34,35,36,37,38,39,40,44,66,68,69,70,71,72,73,74,77,78,83,84,85]; the published cases with specific genetic testing were already introduced in Table 3 [17,53,61,62,64,65,67].

First Authors Reference Number	Year of Publication	Patient	Preoperative Findings	Postoperative Findings and Outcome
Muse [9]	2023	16-year-old male	Nausea, vomiting, headache	Giant PT adenoma This is the first report of a PHP-related brain calcification at the frontal lobe in a child
Sahu [11]	2023	12-year-old female	History of limb deformities, multiple fragility fractures, kidney stones	Ectopic PT adenoma (intra-thymus) Sestamibi-guided thoracoscopic left thymectomy (CT with radioisotope scans)
Boggs [15]	2023	12-year-old female	Chest pain and dyspnea (complication: one osteolytic rib lesion)	Cystic left inferior PT adenoma Hungry bone syndrome following parathyroidectomy
Badhe [83]	2023	13-year-old male	Unilateral slipped capital femoral epiphysis	Ectopic mediastinal PT adenoma
Zenno [84]	2023	9-year-old female	Symptomatic hypercalcemia	Ectopic PT adenoma (pyriform sinus) → reperformed parathyroidectomy
de Silva [40]	2023	17-year-old male	Multiple fractures (brown tumors)	Good outcome after parathyroidectomy
Prakash [76]	2023	14-year-old female	PHP	Lincoln sign (“black beard sign”) due to mandibular uptake of the tracer during 8 F-fluorocholine PE/CT
Gafar [37]	2022	12-year-old male	A 6-year history of bone pain, loss of appetite, fatigue, progressive lower limb deformity	Removal of inferior parathyroid adenoma → postoperative hypocalcemia → calcium and alfacalcidol replacements Association with genu valgum due to rickets
Hayashi [66]	2022	11-year-old male	Hypercalcemic crisis	Poor response to standard care (including calcitonine → pamidronate) → successful emergency parathyroidectomy
Vitale [69]	2022	12-year-old male	Bilateral slipped capital femoral epiphysis	Ectopic PT adenoma (intra-thymus) Thoracoscopic resection → post-parathyroidectomy hungry bone syndrome
Boro [70]	2022	16-year-old female	Severe clinical picture	Atypical PT adenoma → postoperative hungry bone disease
Oh [34]	2022	9-year-old female (C1) 14-year-old male (C2) 14-year-old female (C3)	Abdominal pain due to pancreatitis (C1) Abdominal pain due to ureter stone (C2) Gait disturbance, weakness (C3)	Postoperative PTH normalization (C3: ectopic PT adenoma)
Bin Yahib [29]	2021	13-year-old female	8-month history of recurrent abdominal pain, nausea, vomiting, bone pain	Ectopic PT adenoma (intra-thymus) Thoracoscopic resection
Dikova [38]	2021	12-year-old female (C1) 15-year-old female (C2)	Admission for genu valgum (orthopedic assessment)	C1 associated a local brown tumor C2 associated a local bone cyst
Tuli [36]	2021	16-year-old female (C1) 14-year-old female (C1)	Brown tumor at the heel (C1) Recurrent abdominal pain, emotional lability, asymptomatic nephrolithiasis (C2)	Preoperative cinacalcet (3 months) use (C2) Successful conventional parathyroidectomy in both cases (hungry bone syndrome in C2 that required calcitriol for 2 years)
Flokas [68]	2021	11-year-old female	Unexpected detection of hypercalcemia during admission for peritonsillar cellulitis	Ectopic PT adenoma (intra-thymic) Thoracoscopic thymectomy (5 months after initial presentation)
Fukaya [35]	2021	12-year-old female	Recurrent abdominal pain, macroscopic hematuria	Parathyroidectomy of a large PT adenoma (1.8 g)
Rahimi [73]	2021	15-year-old female	8-year history of bone pain and progressive limping	Parathyroid carcinoma
Roztoczyńska [77]	2020	15-year-old male	Bilateral slipped capital femoral epiphysis, polydipsia, polyuria, weight loss	Orthopedic correction (brown tumors) → medical therapy for hypercalcemia (including pamidronate) → left inferior parathyroidectomy
Legault [71]	2020	14-year-old male	Abdominal pain, vomiting, constipation, pelvic brown tumor	Parathyroidectomy → hungry bone syndrome Postoperative brown tumor remission
Minelli [85]	2020	17-year-old female	Psychiatric manifestations	Ectopic PT adenoma (intra-thymus) Robot-assisted surgery for tumor removal
Seo [10]	2020	15-year-old male	Adenoma localization required SPECT/CT	Ectopic PT adenoma (intra-thymus) → VATS
Lenherr-Taube [72]	2020	13-year-old male	Muscle and bone pain, brown tumors	Parathyroid carcinoma (loss of parafibromin) Therapy with pamidronate, denosumab → parathyroidectomy → hungry bone syndrome (intravenous calcium for 3 weeks) → no relapse for 18 months Postoperative brown tumor remission
Pal [44]	2020	12-year-old male 16-year-old male	Posterior reversible encephalopathy syndrome	Neurological symptom remission after parathyroidectomy
David [74]	2020	15-year-old female	Brown tumors	Parathyroidectomy → hungry bone syndrome
Lee [39]	2020	15-year-old male	Bilateral genu valgum	Parathyroidectomy → calcium normalization within 2 months
Omi [78]	2020	13-year-old female	Fibular fracture	Parathyroidectomy (confirmation of parathyroid carcinoma) → at age of 22: femoral fracture → resection of bilateral lung metastases → at age of 33: re-resection of pulmonary metastases → at age of 57 → suspected neck recurrence → en bloc resection of parathyroid adenoma → ^11^C-methionine-positive tumor recurrence → removal → relapse 8 months later → denosumab for hypercalcemia + radiotherapy to control the recurrence

Abbreviations: C = case; CT = computed tomography; PT = parathyroid; PHP = primary hyperparathyroidism; SPECT/CT = single-photon emission computerized tomography/computed tomography; VATS = video-assisted thoracic surgery.

**Table 5 biomedicines-11-02810-t005:** Case reports and series of neonatal hyperparathyroidism due to inheritance of *CASR* pathogenic variants; PubMed access (between January 2020 and July 2023) [99,101,102,103,104,105,106,107,108].

First Author Reference Number	Year of Publication	Patient	Clinical Presentation	Management
Shaukat [102]	2022	6-month-old male	Lethargy, bradycardia, hyperreflexia (his sibling died of the same condition)	*CASR* pathogenic variant Poor control of hypercalcemia through pamidronate, cinacalcet → parathyroidectomy+ self-transplantation of half of the left inferior PT gland → calcium and alfacalcidol supplements
Özgüç Çömlek [103]	2022	6-day-old male	Hypotonia, poor feeding → weight loss	Therapy with cinacalcet for 13 months → hypercalcemia after 2 months since stopping cinacalcet→ parathyroidectomy
Gupta [104]	2022	20-day-old female	Growth and developmental delays	Parathyroidectomy at 7 months
Hassan [105]	2022	3 infant siblings	Polyuria, failure to thrive, fractures	2/3 patients (a 7-month-old female and an 8-month-old male) were tested and found positive for homozygous missense *CASR* pathogenic variant: c 2038 C T p (Arg680Cys) Patients underwent parathyroidectomies
Höppner [101]	2021	2 infant siblings	Hypercalcemia + preterm (C1) Hypercalcemia + severe muscular hypotonia + thrombocytopenia + multiple fractures (C2)	Heterozygous *CASR* pathogenic variant: c.554G>A; p. (Arg185Gln) Both patients were stabilized under medical therapy (C2: therapy with cinacalcet was used for 3 months)
Gulcan-Kersin [108]	2020	2-day-old female	Severe hypercalcemia during the first day of life	Homozygous *CASR* pathogenic variant: c.1836G>A (p.G613E) Therapy with cinacalcet since day 2 → 18 months (heterozygote status of her father and sister)
Abdullayev [106]	2020	7-month-old male	Multiple episodes of hypercalcemia early after birth	Therapy with calcitonin, cinacalcet, pamidronate → parathyroidectomy twice at 7 months (first removal of all 4 PT glands → persistent high PTH → thymectomy)
Sorapipatcharoen [107]	2020	10-day-old female	Severe hypercalcemia early after birth	Homozygous *CASR* pathogenic variant 1630 (c.1630C > T) Therapy with calcitonin, cinacalcet, pamidronate, zoledronate → parathyroidectomy (day 70) → hungry bone syndrome
Sadacharan [99]	2020	4 cases (mean age of 28.7 days)	Severe hypercalcemia early after birth (hypotonia, respiratory insufficiency, failure to thrive)	Medical therapy → parathyroidectomy + trans-cervical thymectomy (+hemi-thyroidectomy in one case)

Abbreviations: CASR = calcium-sensing receptor, PTH = parathyroid hormone; PT = parathyroid.

## Data Availability

Not applicable.

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
