# Peer review of "Pediatric Neuroendocrine Neoplasia of the Parathyroid Glands: Delving into Primary Hyperparathyroidism"

_biomedicines, 2023, doi:10.3390/biomedicines11102810_

Round 1
Reviewer 1 Report
Dear Authors, congratulations for the submitted work. Here are some elements in this Article I’d like to underline:
-Haven’t you checked for other suitable papers (full-length, English published papers etc.) on other online databasases? If not, why?
-Line 134: in the text is present the typo “clinilca”
-Line 134: The here cited study with a median age of 17 years, may not be fully representative of the characteristics of the disease in the paediatric population, hence a significant proportion of the patients would be already above 18 years old. Which is the upper age bracket you are considering in this review? Also in Table 2 you presented studies with an upper limit of 20 years of age. In my opinion it should be clearly stated the maximum age you used to include or exclude the used studies, and conversely you should use the same age cap about the proposed examples in the papaers (ie line 145: below 22 years old). Please clarify.
-Referring to Chapter 4.4 regarding osteoporosis, there is any data on the role of trabecular bone score (TBS) in these patients?
Author Response
Response to Review 1 Comments
Dear Reviewer,
Thank you very much for your time and your effort to review our manuscript.
We are very grateful for providing your valuable feedback on the article.
Here is our response and related amendment that has been made in the manuscript according to your review (marked in yellow color).
Dear Authors, congratulations for the submitted work.
Thank you very much. We really appreciate it.
Here are some elements in this Article I’d like to underline:
Thank you. We followed each of your points as we introduced them below and we thank you once again.
-Haven’t you checked for other suitable papers (full-length, English published papers etc.) on other online databasases? If not, why?
Thank you very much. Since this is a narrative, not a systematic review, we focused as part of the main research on one database, namely PubMed, which is a well-respected and widely recognized database. This type of approach is a standard one, commonly known, representing a useful tool of overviewing a difficult topic like pediatric primary hyperparathyroidism. However, the observations, the comments and associated data we provided beyond the sample – based analysis from Results section are based on articles that are published in journals indexed in different databases, as seen in Introduction and Discussion.
Moreover, we mentioned this aspect at Discussion, as following:
“We are aware of the limitations of a narrative review mainly focused on PubMed – based research for the sample – based analysis”…
Also, the methods we chosen allowed us to obtain one of the most complexes analyses of this type so far, as mentioned at Discussion due to the heterogeneity of the data that have been published in this challenging topic involving various data belonging to several medical and surgical domains such as pediatric, endocrinology and surgery.
“For instance, Bin Yahib et al. [29] analysed 13 previously published papers (N=331 children, teenagers and young adults) with recurrent abdominal pain as feature of PHP [29]. Lee et al. [39] studied prior reports of young patients with genu valgum and PHP and identified 23 cases [39]. Rahimi et al. [73] analysed 17 previously published subjects with regard to paediatric parathyroid carcinoma [73], another review identified 12 individuals confirmed with the same histological diagnosis and added one new report (N=13) [72]. In 2021, Flokas et al. [68] published a review of 33 ectopic parathyroid juvenile adenomas [68]. Due the level of statistical evidence concerning the area of paediatric PHP publications, our study – and case – sample – based analysis that included 48 articles, respectively, a total of 534 published cases of PHP represents one of the largest of its kind.”
Thank you
-Line 134: in the text is present the typo “clinilca”
Thank you very much. We corrected it. Thank you.
-Line 134: The here cited study with a median age of 17 years, may not be fully representative of the characteristics of the disease in the paediatric population, hence a significant proportion of the patients would be already above 18 years old. Which is the upper age bracket you are considering in this review? Also in Table 2 you presented studies with an upper limit of 20 years of age. In my opinion it should be clearly stated the maximum age you used to include or exclude the used studies, and conversely you should use the same age cap about the proposed examples in the papaers (ie line 145: below 22 years old). Please clarify.
Thank you very much. As mentioned at Methods section, we looked for studies on “pediatric” population as named by the original authors. This means persons younger than 18 years old at the moment of primary hyperparathyroidism diagnosis (not necessarily the parathyroid surgery), but this diagnosis does not mean that the patient, for instance, does not have a prior medical history or follow-up data with respect to long-term period.
Overall, the included studies had a mean or median below 18 years. We agree with your point, this is one of raisons we did not limit the research according to a systematic review in order to have a more flexible approach.
Moreover, the issue of clearly cutting the data is related to the fact that many centers address young adults (for instance, of 18-year-old) as well as teenagers (for instance, of 15-year-old) that is why some studies cannot be completely overlooked.
However, we specified the point you mentioned according to the issue of specific patients’ age and we discussed the subjects’ age aspects.
“The admission of the patients within the mentioned study designs were between 1989 and 2021 (different time frames combinations). The mean age of the patients was of 10.2 (the lowest mean value), around 14 years (for example, of 14, 14.1, 14.7, and 15.2 years in other studies), and around the age of 17 (for instance, of 16.7, 17, and 17.3) [5,18,26-28,45-52]. We exceptionally included 4 studies that applied as (maximum) cut off age of 20-21 (and not 18), but the average and median ages of the entire studied populations were below 18 [5,26,27,52]. Case reports included the youngest child of 9 years, respectively the oldest of 17 [9,10,11,15,29,34-40,44,66,68-74,77-78,83-85].”
Thank you
-Referring to Chapter 4.4 regarding osteoporosis, there is any data on the role of trabecular bone score (TBS) in these patients?
Thank you very much. We are not aware of such data according to our methods, but, indeed we agree that this is going to be a wonderful topic to explore and new results are expected for TBS – related values with regard to primary hyperparathyroidism in children and teenagers.
We commented this aspect at Discussion:
“Lately, lumbar DXA – based trabecular bone score (TBS), as practical tool for bone microarchitecture assessment, was applied in individuals younger than 18 years to a lesser extent than adults, but the topic continues to expand and we anticipate valuable paediatric studies in PHP, as well [119,120].”
Additionally, we cited two studies on healthy children, not with primary hyperparathyroidism:
“Fraga MM, de Sousa FP, Szejnfeld VL, de Moura Castro CH, de Medeiros Pinheiro M, Terreri MT. Trabecular bone score (TBS) and bone mineral density (BMD) analysis by dual X-ray absorptiometry (DXA) in healthy Brazilian children and adolescents: normative data. Arch Osteoporos. 2023;18(1):82. doi:10.1007/s11657-023-01291-1.
Valenzuela Riveros LF, Long J, Bachrach LK, Leonard MB, Kent K. Trabecular Bone Score (TBS) Varies with Correction for Tissue Thickness Versus Body Mass Index: Implications When Using Pediatric Reference Norms. J Bone Miner Res. 2023;38(4):493-498. doi:10.1002/jbmr.4786.”
Thank you
Thank you very much.

Reviewer 2 Report
This is a review paper analyzing recent data concerning primary 19 hyperparathyroidism (PHP) in children and teenagers. The manuscript is written in fine English and does not need significant language editing.
The title should be amended to reflect the type of study and content, according to PRISMA guidelines.
The abstract is too long, difficult to follow and presents too much data. I suggest completely rewriting it for clarity, following the usual IMRaD organization.
The objectives of the study are presented clearly and the introduction section communicates the need for assessing diagnostics of pediatric PHP in a multidisciplinary way, alongside therapeutic outcomes. The clinical picture and pathophysiology sections have outlined valuable differences regarding PHP presentation in the young.
In the MM section, PRISMA guidelines need to be followed and cited.
The paper did a good job on presenting the major etiologic, genetic and clinical implications of PHD, while discussing relevant co-morbidity. However, it needs to be edited for clarity and flow. The text often times describes results duplicated in the tables, and should be shortened. In addition, singular case reports occupy too much space in the narrative. Although interesting, the point of the review is to systematically update, not further complicate diagnostics with rare PHP presentations. Again, the same is true with sections dealing with treatment, with small volumes of cases dealt with through thoracoscopic approaches dominating the discussion.
The images and tables are easy to read, sufficient in both number and detail to enable a comprehensive addition to the information detailed in the text. I would suggest transferring all non-vital and case-report-level information into tables, and condensing the text wherever possible.
The scope and breadth of the discussion is more than appropriate for a detailed review.
Proofreading by a native speaker necessary.
Author Response
Response to Review 2 Comments
Dear Reviewer,
Thank you very much for your time and your effort to review our manuscript.
We are very grateful for your insightful comments and observations, also, for providing your valuable feedback on the article.
Here is a point-by-point response and related amendments that have been made in the manuscript according to your review (marked in yellow color).
This is a review paper analyzing recent data concerning primary 19 hyperparathyroidism (PHP) in children and teenagers. The manuscript is written in fine English and does not need significant language editing.
Thank you very much. We really appreciate it.
The title should be amended to reflect the type of study and content, according to PRISMA guidelines.
Thank you very much. PRISMA stands for “Preferred Reporting Items for Systematic Reviews and Meta-Analyses” (http://prisma-statement.org/prismastatement/flowdiagram.aspx) and this is a narrative review, not a systematic one. Due to the heterogeneity of the spectrum in primary hyperparathyroidism with regard to pediatric population, we choose to introduce the data as a narrative review since various levels of statistical evidence are identified in mentioned papers. On the other hand, a systematic review pinpoints a specific critical assessment which in the matter of juvenile primary hyperparathyroidism is rather limited so far.
However, this type of review is a well-recognized, standard, traditional approach which is suitable for topics with less generous publications such as the update of the most recent data on pediatric primary hyperparathyroidism. This allowed us to examine and evaluate the scientific panel on this specific topic in a useful way for various practitioners.
We mentioned at Discussion the aspect of this research/paper approach.
“We are aware of the limitations of a narrative review…”
Thank you
The abstract is too long, difficult to follow and presents too much data. I suggest completely rewriting it for clarity, following the usual IMRaD organization.
Thank you. We reduced the length and reorganized IMRAD sections are followed according to our type of article. Thank you
The objectives of the study are presented clearly and the introduction section communicates the need for assessing diagnostics of pediatric PHP in a multidisciplinary way, alongside therapeutic outcomes. The clinical picture and pathophysiology sections have outlined valuable differences regarding PHP presentation in the young.
Thank you. We really appreciate it.
In the MM section, PRISMA guidelines need to be followed and cited.
Thank you very much. We already answered above to this aspect. Thank you.
The paper did a good job on presenting the major etiologic, genetic and clinical implications of PHD, while discussing relevant co-morbidity. However, it needs to be edited for clarity and flow. The text often times describes results duplicated in the tables, and should be shortened. In addition, singular case reports occupy too much space in the narrative. Although interesting, the point of the review is to systematically update, not further complicate diagnostics with rare PHP presentations. Again, the same is true with sections dealing with treatment, with small volumes of cases dealt with through thoracoscopic approaches dominating the discussion.
Thank you very much. The point with case reports is that some aspects are only at this level of statistical evidence and we wanted to highlight some unexpected clues, unexpected outcomes or hormonal/imaging panels for specialists that are not necessarily experts in the field of primary hyperparathyroidism. This is meant to increase the awareness and to facilitate an early disease recognition and management. The fact that we resonated, as well, with the case reports is based on practical and meaningful key messages rather than statistical evidence. According to your suggestions we reduced the data of some cases.
Thoracoscopic approach represents one of the modern aspects in ectopic parathyroid pediatric adenomas. For a very long time this technique was considered not suitable and not safe for children opposite to adults diagnosed with primary hyperparathyroidism caused by ectopic parathyroid tumors, that is why we intended to capture the extension of this novel procedure. As we mentioned, the level of statistical evidence with concern of this particular matter still remains low, but the data are new and we expect an expansion and progress of this domain.
Also, as we mentioned, we intended to provide a complex overview of pediatric hyperparathyroidism from a multi-disciplinary perspective, including surgical perspective, approaches, intra-operatory assessments, post-operatory outcomes, unexpected pitfalls, timing of surgery, cure rate, etc.
Thank you
The images and tables are easy to read, sufficient in both number and detail to enable a comprehensive addition to the information detailed in the text. I would suggest transferring all non-vital and case-report-level information into tables, and condensing the text wherever possible.
Thank you very much. Since these data are massive and very different, and complex, we experienced different types of tables until the present ones and finally decided to these tables format and associated data that we respectfully choose to keep. We already have 5 complex tables and this article is meant to be a narrative update that is why we respectfully opt for this presentation. Thank you very much.
The scope and breadth of the discussion is more than appropriate for a detailed review.
Thank you very much. We really appreciate it.
Comments on the Quality of English Language: Proofreading by a native speaker necessary.
Thank you. A correction was done. Thank you
Thank you very much,

Reviewer 3 Report
The article focuses on neuroendocrine tumors causing pseudohypoparathyroidism (PHP) in children and teenagers. It aims to provide an overview of the most recent data regarding these tumors from a multidisciplinary perspective, including pathogenic traits, clinical presentation, genetic analysis, management, and outcomes. The authors conducted a narrative review, searching for full-length, English-published papers that were freely accessible via PubMed. The inclusion criteria included studying the population of children and teenagers (patients of 18 years old or younger) and having research keywords within the title and/or abstract. Exclusion criteria and specific methods regarding the literature search are also mentioned, though these are not entirely clear from the extracted text.
Following are the limitations that need to be addressed.
1. The study is a narrative review, which may not provide the same level of evidence as a systematic review or meta-analysis.
2. The search for relevant articles was limited to full-length, English-published papers that were freely accessed via PubMed. This may have excluded relevant studies published in other languages or not indexed in PubMed.
3. The search was focused on specific keywords ("pediatric primary hyperparathyroidism," "child and primary hyperparathyroidism," and "teenager and primary hyperparathyroidism"), which may have missed relevant articles using different terminology.
4. The study excluded certain articles, such as reviews, editorials, and experimental studies, which may have provided valuable insights or context.
5. The time frame for the search was limited to publications between January 2020 and July 2023, potentially excluding earlier studies that could have contributed to the understanding of the topic.
good
Author Response
Response to Review 3 Comments
Dear Reviewer,
Thank you very much for your time and your effort to review our manuscript.
We are very grateful for providing your valuable feedback on the article.
Here is our response and related amendment that has been made in the manuscript according to your review (marked in yellow color).
The article focuses on neuroendocrine tumors causing pseudohypoparathyroidism (PHP) in children and teenagers. It aims to provide an overview of the most recent data regarding these tumors from a multidisciplinary perspective, including pathogenic traits, clinical presentation, genetic analysis, management, and outcomes. The authors conducted a narrative review, searching for full-length, English-published papers that were freely accessible via PubMed. The inclusion criteria included studying the population of children and teenagers (patients of 18 years old or younger) and having research keywords within the title and/or abstract. Exclusion criteria and specific methods regarding the literature search are also mentioned, though these are not entirely clear from the extracted text.
Thank you very much. We really appreciate it.
Following are the limitations that need to be addressed. The study is a narrative review, which may not provide the same level of evidence as a systematic review or meta-analysis.
Thank you. Indeed, this is a narrative review, not a systematic one. Due to the heterogeneity of the spectrum in primary hyperparathyroidism with regard to pediatric population, we choose to introduce the data as a narrative review since various levels of statistical evidence are identified in mentioned papers. On the other hand, a systematic review pinpoints a specific critical assessment which in the matter of juvenile primary hyperparathyroidism is rather limited so far. However, this type of review is a well-recognized, standard, traditional approach which is suitable for topics with less generous publications such as the update of the most recent data on pediatric primary hyperparathyroidism. This allowed us to examine and evaluate the scientific panel on this specific topic in a useful way for various practitioners. We mentioned at Discussion the aspect of this research/paper approach. “We are aware of the limitations of a narrative review”.Thank you.
The search for relevant articles was limited to full-length, English-published papers that were freely accessed via PubMed. This may have excluded relevant studies published in other languages or not indexed in PubMed.
Thank you. The non-English language involves a translation which is a traditional bias that we respectfully chosen to avoid. However the studied population is from all over the word. We highlighted some geographical aspects in interesting articles, for instance:
“The first such report from Sudan included 3 siblings carrying a homozygote missense CASR pathogenic variant and, since medical alternatives were not available, parathyroidectomy was performed..”
“Similarly, genetic/familial PHP versus apparently sporadic PHP was studied according to a single-center analysis published in 2022 by Sharma et al. [26] (N=36 Asian Indian patients younger than 20 years; 55% were males; median age of 17 years). 90% of the subjects from the first group mostly carried MEN1 and CDC73 pathogenic variants.”
“Rickets may be co-incidental, sometimes this delays the recognition of PHP. For example, a 12-year-old boy coming from Sudan was identified with a 6-month history of bone pain, loss of appetite, fatigue in associated with progressive lower limb deformity that impaired the normal walking.”
Examples of references:
Sharma A, Memon S, Lila AR, Sarathi V, Arya S, Jadhav SS, Hira P, Garale M, Gosavi V, Karlekar M, Patil V, Bandgar T. Genotype-Phenotype Correlations in Asian Indian Children and Adolescents with Primary Hyperparathyroidism. Calcif Tissue Int. 2022;111(3):229-241. doi:10.1007/s00223-022-00985-x.
Sharanappa V, Mishra A, Bhatia V, Mayilvagnan S, Chand G, Agarwal G, Agarwal A, Mishra SK. Pediatric Primary Hyperparathyroidism: Experience in a Tertiary Care Referral Center in a Developing Country Over Three Decades. World J Surg. 2021;45(2):488-495. doi:10.1007/s00268-020-05816-4.
Gafar SM, Fadlalbari GF, Abdalla AT, Mohammed SAR, Alrasheed MK, Taha IA, Abdullah MA. Pitfalls in the Diagnosis of Primary Hyperparathyroidism in a Sudanese Adolescent Boy; a case disguised as rickets. BMC Endocr Disord. 2022;22(1):322. doi:10.1186/s12902-022-01241-x.
Moreover, we focused on the main research on one database, namely PubMed, which is a well-respected and widely recognized database. This type of approach is a standard one, commonly known, representing a useful tool of overviewing a difficult topic like pediatric primary hyperparathyroidism. However, the observations, the comments and associated data we provided beyond the sample – based analysis from Results section are based on articles that are published in journals indexed in different databases, as seen in Introduction and Discussion sections. We mentioned this aspect at Discussion, as following: “We are aware of the limitations of a narrative review mainly focused on PubMed – based research for the sample – based analysis”. Thank you
The search was focused on specific keywords ("pediatric primary hyperparathyroidism," "child and primary hyperparathyroidism," and "teenager and primary hyperparathyroidism"), which may have missed relevant articles using different terminology.
Thank you very much. We searched other combinations and received no additional information, that is why we introduced the mentioned combinations which are the most relevant, the most efficient and the most accurate. Thank you.
The study excluded certain articles, such as reviews, editorials, and experimental studies, which may have provided valuable insights or context.
Thank you very much. Only the sample – based analysis was focused on original studies, but, as you can check in our reference list, we cited others reviews and editorials at Introduction and Discussion sections. Since these are clinical aspects with practical points, experimental studies are out of the scope. Of note, at Discussion, we pinpointed new data, further frontiers to explore, off label use of drugs such as cinacalcet or denosumab, etc. to bridge the gap to experimental data. Thank you.
The time frame for the search was limited to publications between January 2020 and July 2023, potentially excluding earlier studies that could have contributed to the understanding of the topic.
Thank you very much. We specifically address this time window to provide an update of the most recent data in pediatric primary hyperaparathyrodism, but we mentioned the most important aspects in relationship with prior publications.
Also, at Discussion section, we explained the importance of such time frame by potential interference of the pandemic years. “Our analysis covered the COVID-19 pandemic era; upon this period of time many medical and surgical conditions have been re-shaped or were associated with new entities due to the coronavirus infection or to the new regulations. With respect to PHP, a delay of hospital admission and potentially a more frequent indication of using calcium lowering agents such cinacalcet or denosumb to function as a bridge to the surgical procedure (if it was not considered an emergency) were registered amid the first waves. However, we found no pediatric study on this specific matter. Moreover, compresive neck symptoms should take into consideration thyroid and parathyroid tumors, but also infectious – asssociated clinical picture….“
Also, we compared our work to others with different outcomes, but in the field of paediatric hyperparathyroidism: ..”we identified a few reviews with different objectives in paediatric PHP, as mentioned. For instance, Bin Yahib et al. [29] analysed 13 previously published papers (N=331 children, teenagers and young adults) with recurrent abdominal pain as feature of PHP [29]. Lee et al. [39] studied prior reports of young patients with genu valgum and PHP and identified 23 cases [39]. Rahimi et al. [73] analysed 17 previously published subjects with regard to paediatric parathyroid carcinoma [73], another review identified 12 individuals confirmed with the same histological diagnosis and added one new report (N=13) [72]. In 2021, Flokas et al. [68] published a review of 33 ectopic parathyroid juvenile adenomas [68].” Overall, the meticulous aspect we have taken into considerations made this work one of the most complexes of its kind. Thank you
Comments on the Quality of English Language: good
Thank you
Thank you very much.

Round 2
Reviewer 2 Report
The revised version of the paper is as close as possible to the final version as we will get, and in light of the effort and content amendments, I have no further issues.